# Fecal Short-Chain Fatty Acids Are Associated with Obesity in Gestational Diabetes

**DOI:** 10.3390/biomedicines13020387

**Published:** 2025-02-06

**Authors:** Katja Molan, Jerneja Ambrožič Avguštin, Matevž Likar, Drazenka Pongrac Barlovic, Darja Žgur Bertok, Marjanca Starčič Erjavec

**Affiliations:** 1Faculty of Health Sciences, University of Novo mesto, 8000 Novo mesto, Slovenia; katja.molan@uni-nm.si; 2Department of Biology, Biotechnical Faculty, University of Ljubljana, 1000 Ljubljana, Slovenia; jerneja.ambrozic@bf.uni-lj.si (J.A.A.); matevz.likar@bf.uni-lj.si (M.L.); darja.zgur@bf.uni-lj.si (D.Ž.B.); 3Department of Endocrinology, Diabetes and Metabolic Diseases, University Medical Centre, 1000 Ljubljana, Slovenia; drazenka.pongrac@gmail.com; 4Faculty of Medicine, University of Ljubljana, 1000 Ljubljana, Slovenia; 5Department of Microbiology, Biotechnical Faculty, University of Ljubljana, 1000 Ljubljana, Slovenia; 6Department of Biology, Faculty of Natural Sciences and Mathematics, University of Maribor, 2000 Maribor, Slovenia

**Keywords:** fecal SCFA, pregnancy, gestational diabetes mellitus, GDM, preconception BMI, gestational weight gain, GWG

## Abstract

**Background:** Short-chain fatty acids (SCFAs), which are produced by the microbial fermentation of undigested carbohydrates, play an important role in the metabolism and physiology of the host. SCFAs are involved in the regulation of maternal metabolism during pregnancy and influence weight gain, glucose metabolism, and metabolic hormones. **Methods:** In 2017, women who were treated for gestational diabetes mellitus (GDM) at the University Medical Centre Ljubljana were invited to participate in a longitudinal study. A total of 45 women were included in this study and comprehensively phenotyped. During the second and third trimester of pregnancy, the women with GDM provided fecal samples for SCFA analysis. The samples were analyzed by high-performance liquid chromatography for the simultaneous determination of acetate, propionate, and butyrate. **Results:** SCFA concentrations in feces differed between overweight/obese and normal-weight women with GDM. Acetate and propionate concentrations were significantly higher in pregnant women who were overweight or obese before pregnancy compared to normal-weight women but butyrate concentrations were not. Butyrate was elevated in the third trimester in the group with excessive gestational weight gain. **Conclusions:** The relationship between SCFAs and obesity is complex, and the association between SCFAs and GDM remains to be clarified. Regardless of the conflicting publications on the role of SCFAs, our study showed that higher acetate and propionate levels were associated with the weight categories of overweight or obesity before pregnancy and higher butyrate levels were associated with excessive gestational weight gain.

## 1. Introduction

Short-chain fatty acids (SCFAs) are the end products of the microbial fermentation of indigestible carbohydrates (dietary fiber) [1]. They are mainly produced in the large intestine, while food-derived SCFAs are present in the small intestine. Thus, the content of SCFAs in the large intestine reflects the combination of food intake and the formation of SCFAs in situ. The most important producers of SCFAs are members of the phylum *Bacteroidetes*, which mainly produce acetate and propionate, and members of the families *Ruminococcaceae* and *Lachnospiraceae* of the phylum *Firmicutes*, which produce butyrate [2]. Genomic profiles of gut bacteria indicate that members of the phyla *Actinobacteria*, *Bacteroidetes*, *Fusobacteria*, *Proteobacteria*, *Spirochaetes*, and *Thermotogae* are also potential butyrate producers [3]. The most important SCFAs are acetate, propionate, and butyrate. They have been shown to have important signaling functions that affect host metabolism and physiology [4,5,6]. Through the epigenetic regulation of gene transcription, SCFAs can influence various cellular processes. This modulation of gene expression is thought to be one of the mechanisms through which SCFAs exert their beneficial effects on host health [7,8].

Various pathologies such as metabolic syndrome, obesity, hyperglycemia, insulin resistance, and gestational diabetes have been linked with the human microbiome [9,10,11]. Different interpretations have been proposed for the role of the microbiota in the mechanisms behind these pathologies. For example, altered concentrations of SCFAs in the feces have been found in obese individuals [12,13]. The microbiota is thought to induce obesity through the fermentation of dietary fiber and overproduction of SCFAs [11,14]. The imbalance in the gut microbiota can result in altered SCFA levels leading to chronic inflammation and metabolic dysfunction [14,15,16,17]. A decline in SCFA-producing bacteria, notably butyrate-producers like *Faecalibacterium prausnitzii* and *Roseburia hominis*, characterizes IBD patients [18,19,20]. In addition, studies have shown that people with type 2 diabetes (T2D) have reduced levels of SCFA-producing bacteria. This is associated with insulin resistance and the progression of T2D and may contribute to inflammation in the gut [18,21,22]. Given the peculiarities of the gut microbiota in patients with T2D and the resulting consequences, probiotics are recognized as beneficial and complementary, especially *Lactobacilli* and *Bifidobacteria*, which have recently gained attention as promising biotherapeutics with proven efficacy in various experimental models [22]. The mechanisms, applications, development prospects, and challenges of probiotics were discussed broadly in a recent review [23], which also provided theoretical references and a guiding basis for the development of probiotic preparations and related functional foods that regulate blood glucose. While probiotics can influence SCFA production and potentially impact obesity, the exact effects depend on various factors, including the specific probiotic strains and individual gut microbiota composition; therefore, more research is needed to fully understand this relationship [24].

Gestational diabetes mellitus (GDM) is defined as any glucose intolerance that first appears in pregnancy [25]. The prevalence varies between 1 and 14 percent of all pregnancies, depending on the population and the diagnostic test used [26]. In total, 35 to 60 percent of women with GDM develop type 2 diabetes within 5 to 10 years after giving birth. In addition, they have a higher risk of complications during pregnancy, and their children are exposed to a higher risk of obesity and type 2 diabetes during adolescence and adulthood [26,27,28,29]. In the majority of GDM cases, the pancreatic β-cells are unable to compensate for the chronic excess fuel, which eventually leads to insulin resistance, hyperglycemia, and increased glucose supply to the growing fetus [30]. GDM is associated with insulin resistance, inflammation, and gut microbiota dysbiosis, all of which can be influenced by SCFAs [31,32]. SCFA-producing genera—including *Faecalibacterium*, *Ruminoccus*, *Bifidobacterium*, and *Akkermansia*—were found to be reduced in individuals with GDM [33,34].

It is known that SCFAs in the mother’s serum influence the metabolic changes that occur during pregnancy, including glucose metabolism, maternal weight gain, and the levels of various metabolic hormones [35,36,37]. SCFA concentrations in the serum were associated positively with maternal adiponectin levels, serum propionate correlated negatively with maternal leptin, and no associations were observed with serum butyrate [36]. A study by Fuller and colleagues [38] found that FFA2 (free fatty acid receptor) signaling is essential for β-cell adaptation in pregnancy, indicating that gut microbiota, SCFA, and FFA2 work in balance during pregnancy to mediate glucose homeostasis. Based on the results, they suggested that there may be a beneficial effect of SCFAs on metabolic health during pregnancy, particularly in overweight and obese pregnant women. Further, it was shown that the level of propionic acid decreases with the course of pregnancy, although for obese pregnant women, it is characteristic that it increases. The increase in propionic acid levels in obese pregnant women and the correlation with glycosylated hemoglobin and fasting glucose levels could be considered a likely factor in glucose metabolism disorders in obese pregnant women [39].

Gomez-Arango et al. [40] reported that SCFA production by gut bacteria was negatively associated with body mass index, meaning that higher SCFA concentrations resulted in lower body mass index. Their results also suggested that an increased number of butyrate-producing bacteria may contribute to lower blood pressure in overweight and obese pregnant women. Changes in isobutyrate, isovalerate, and 2-methylbutyrate were inversely related to lower blood pressure, and increased valerate levels were associated with reduced carotid–femoral pulse wave velocity and, consequently, a lower risk of cardiovascular complications [41,42]. Ecklu-Mensah et al. [43] reported that obesity was significantly associated with a reduction in SCFA concentrations, microbial diversity, and SCFA-synthesizing bacteria, and that the country of origin was the strongest explanatory factor for differences observed in the study.

While SCFAs are an important source of energy for colonocytes and can also regulate host metabolism, excessive SCFA production and uptake may contribute to obesity and related metabolic disorders. SCFAs can be oxidized for energy, and their metabolites can be used for de novo lipid synthesis and gluconeogenesis [44]. A meta-analysis has shown that among seven studies evaluated, people with obesity had significantly higher fecal SCFA content compared to lean controls [45]. Excessive lipid synthesis and storage can contribute to the development of obesity, particularly when combined with a high-calorie diet and a sedentary lifestyle [46]. De la Cuesta-Zuluaga [47] reported that higher SCFA levels in the feces were associated with indicators of lower gut microbiota diversity, and higher gut permeability, systemic inflammation, glycemia, dyslipidemia, obesity, and hypertension and they suggested that greater excretion of SCFAs in feces is a marker of poor gut health and cardiometabolic dysregulation.

The relationship between SCFA and body weight is complex and not fully understood. There are no data on the concentration or ratio of fecal SCFA in pregnant women with gestational diabetes. Therefore, the aim of this study was to determine the concentrations of SCFA in the feces of pregnant women with GDM in the second and third trimesters of pregnancy. In this study, we measured the concentrations of acetate, propionate, and butyrate. We hypothesized that the fecal SCFA concentrations in overweight/obese women with GDM would differ from the fecal SCFA concentrations in normal-weight women with GDM in the second and third trimesters of pregnancy.

## 2. Materials and Methods

### 2.1. Women’s Inclusion/Recruitment and Fecal Sample Collection

Women who were treated for GDM in an outpatient clinic at the University Medical Center Ljubljana in 2017 were invited to participate in our longitudinal study (approved by the Slovenian Medical Ethics Committee, No. 0120–323/2016–2 on 16 November 2016). The diagnosis of GDM was based on an OGTT test. The exclusion criterion was at least six months without antibiotic therapy.

Out of 57 recruited women with GDM, 45 provided fecal samples in both tested trimesters, the 2nd and 3rd trimesters. In both trimesters of pregnancy, the women with GDM were given a fecal collection kit with detailed instructions. The fecal samples collected were immediately stored in home refrigerators at –20 °C and within 24 h collected and transferred on ice to the laboratory where the samples were aliquoted and stored at –80 °C until extraction of the SCFA. In total, 45 women with GDM were thoroughly phenotyped (age, Body Mass Index (BMI) preconceptionally, gestational weight gain, insulin, glycated hemoglobin, blood pressure).

### 2.2. Extraction of SCFAs

A total of 200 mg of feces was resuspended in 1 mL of 70% ethanol (Merck KGaA, Darmstadt, Germany), vortexed, and then centrifuged at 2500 rpm for 10 min at 20 °C. The supernatant was transferred to a new Eppendorf tube and stored at –80 °C until further use, i.e., derivatization.

### 2.3. Derivatization and HPLC Analysis

Derivatization was performed as previously described [48]. Briefly, 300 µL of each sample was mixed with 300 µL of reagent 1 (pyridine, Merck KGaA, Darmstadt, Germany), reagent 2 (EDC-HCl, Merck KGaA, Darmstadt, Germany), and reagent 3 (2-NPH-HCl, Merck KGaA, Darmstadt, Germany) and incubated at 60 °C for 20 min. Then, 200 µL of reagent 4 (potassium hydroxide solution) was added. The mixture was shaken and allowed to react at 60 °C for 20 min. After the mixture was cooled, 3 mL of aqueous phosphoric acid solution (0.5 M) and 4 mL of ether (Merck KGaA, Darmstadt, Germany) were added and shaken for 3 min, followed by centrifugation for 15 min (2500 rpm, 20 °C). The upper ether layer was transferred and mixed with 4 mL of MQ water and centrifuged (15 min, 2500 rpm, 20 °C). Subsequently, 1 mL of the ether layer was transferred to a glass vial. The ether was eliminated with N_2_. The obtained fatty acid hydrazide was dissolved in 1 mL methanol and 30 µL was subjected to HPLC.

HPLC was performed for simultaneous determination of acetate, propionate, and butyrate. The analyses were performed using an HPLC system (2965 HPLC system; Waters, Milford, MA, USA), a photodiode array detector (237 nm), and a fluorescence detector (excitation, 331 nm; emission, 500 nm). A C18 HPLC column was used (150 mm × 4.6 mm, particle size 5 µm; Kinetex XB; Phenomenex, Torrance, CA, USA). The flow rate of the mobile phase was 1 mL/min and the injection volume was 30 µL. The separation of SCFAs was carried out at 30 °C using two eluents: A, 0.1% trifluoroacetic acid in ultra-pure water; B, acetonitrile. A discontinuous gradient was run as follows: 0–0.5 min, 30% B; 0.5–2.0 min, 30–50% B; 2.0–14.0 min, 50% B; 14.0–17.5 min, 50–70% B; 17.5–20.5 min, 70% B; 20.5–22.5 min, 70–100% B; 22.5–62.5 min, 100% B; 62.5–63.0 min, 100–30% B; 63.0–65.0 min, 30% B. The retention times were identified by injection of acetate, propionate, and butyrate standards (Merck KGaA, Darmstadt, Germany). The peaks of the compounds analyzed in the samples were also confirmed by spiking the samples with commercial standards.

### 2.4. Statistical Analysis

Statistical analyses were performed using IBM SPSS Statistics version 26.0. Normality for each data set was evaluated with the Kolmogorov–Smirnov test. The mean ± standard deviation was given for normally distributed data, whereas the median value (and 25th and 75th percentiles) was given for non-normally distributed data. Differences in fecal SCFA content among groups were analyzed by the ANOVA test. The correlations between individual parameters were analyzed using Pearson’s matrices. A *p*-value less than 0.05 was considered statistically significant.

## 3. Results

### 3.1. Baseline Characteristics of Women with Gestational Diabetes Mellitus

A total of 45 women provided fecal samples in their second and third trimesters of pregnancy. As can be seen in Table 1, the women were divided into the groups normal-weight (32 women) and overweight/obese (13 women) according to their BMI before conception (*p* < 0.001). Overweight/obese women gained significantly less weight during pregnancy (GWG) than women with a normal BMI before pregnancy (*p* = 0.017). At the time of inclusion in this study (second trimester), insulin levels differed significantly between the groups (*p* = 0.002) as well as serum gamma-glutamyl transferase concentrations (0.006), while serum alanine transaminase (S-ALT), serum aspartate aminotransferase (S-AST), cholesterol, and triglycerides were comparable between the groups. In the second trimester, blood pressure was higher in the overweight/obese group than in the normal-weight group, although not significantly. Glycated hemoglobin did not differ between the groups and was stable during pregnancy [49].

The heat map of the correlation between SCFAs and other variables is shown in Figure 1. All major SCFAs (acetate, propionate, and butyrate) correlated positively with each other in the second and third trimesters. Acetate and propionate concentrations correlated positively with BMIp but this was significant only in the third trimester. Acetate concentrations in the second trimester correlated significantly (positive) with second-trimester serum cholesterol and serum LDL cholesterol. A positive correlation between acetate concentration and diastolic blood pressure (BP) was observed in both trimesters, although a significant correlation between the parameters was only observed in the third trimester. A significant (negative) correlation was observed between the second-trimester butyrate concentrations and the second-trimester glycated hemoglobin values. A negative correlation was also observed between the third-trimester butyrate levels and third-trimester glycated hemoglobin but the correlation was not significant. Glycated hemoglobin from the second trimester correlated significantly (positive) with serum gamma-GT concentrations and negatively with cholesterol and triglyceride concentrations. A significant negative correlation between gestational weight gain (GWG) and glycated hemoglobin in the third trimester was observed. Between GWG and second-trimester serum HDL cholesterol concentrations, a significant positive correlation was seen. Second-trimester serum gamma-GT correlated significantly (negatively) with the second-trimester cholesterol values and significantly (positively) with the second-trimester glycated hemoglobin. Second-trimester serum cholesterol concentrations and triglycerides correlated significantly (negatively) with the second-trimester glycated hemoglobin concentrations. Second-trimester insulin concentrations correlated significantly (positively) with BMIp and serum triglycerides.

### 3.2. SCFAs in Relation to BMI Preconceptionaly

Two women in the second trimester (one woman from the normal-weight group and one from the overweight/obese group) and two in the third trimester (both women from the normal-weight group) did not provide a sufficient amount of feces for SCFA analysis. Therefore, of 45 women with GDM, the SCFAs of 43 women with GDM from the second trimester and 43 women with GDM from the third trimester were analyzed. As presented in Table 2, in total, fecal SCFAs (acetate, propionate, and butyrate) differed significantly (*p* < 0.05) between the tested groups in both trimesters of pregnancy, with higher concentrations in the overweight/obese group. When analyzing SCFAs separately, acetate and propionate differed significantly between the groups in both tested trimesters but butyrate did not. SCFA concentrations did not differ significantly between the second and third trimester of pregnancy. The acetate–propionate–butyrate ratio was approximately 65:10:25 in both of the tested trimesters.

### 3.3. Butyrate and Excessive Gestational Weight Gain (GWG)

A total of 7 women with GDM gained excessive weight (gestational weight gain exceeding recommendations, excessive GWG), and 35 women gained a normal amount of weight during pregnancy (Figure 2a). Groups were determined according to the Institute of Medicine (IOM) GWG guidelines [50]. For two women with GDM with normal GWG, no data on the butyrate concentration in the third trimester were available, and for three women with GDM, no data on the GWG were available. Women with excessive gestational weight gain (7 women with GDM) in the third trimester of pregnancy had significantly higher concentrations of fecal butyrate when compared to women with normal gestational weight gain (33 women with GDM) (*p* = 0.037) but not acetate and propionate (Figure 2b).

## 4. Discussion

Due to their importance, SCFAs are the subject of many studies; however, they are often not comparable due to different methodologies, e.g., SCFA concentrations can be reported per dry or wet weight of feces or as mM of SCFAs in fecal water. Here, we report for the first time the concentrations of acetate, propionate, and butyrate in the feces of women with GDM. As the relationship between SCFAs and body weight is not yet fully understood, we measured the concentrations of the three major SCFAs and hypothesized that fecal SCFA concentrations would differ between overweight/obese and normal-weight pregnant women.

To the best of our knowledge, only three studies have been published that quantified SCFAs from the feces of pregnant women [39,51,52]. Jost and colleagues [51] found that SCFA concentrations in the feces of pregnant women were elevated compared to the literature data that included SCFA analyses in non-pregnant individuals. Increased SCFA concentrations during pregnancy are thought to reflect metabolic changes and low levels of inflammation, which are considered to be beneficial in the context of a healthy pregnancy [35,36,51]. This may be consistent with our findings as we measured relatively high concentrations of SCFAs (µmol/g feces, wet weight) in the feces of pregnant women with GDM, which were higher than those reported in the literature for the healthy human population, including non-pregnant women [12,53,54].

Høverstad et al. [53] reported SCFA concentrations per wet weight of feces (mmol/kg feces = µmol/g feces) as 37.4 for acetate, 12.5 for propionate, and 12.4 for butyrate. Fernandes and colleagues [54] also showed similar SCFA values for the wet weight of feces in mmol/kg. The concentrations in our research (311.0 for acetate, 48.7 for propionate, and 120.5 for butyrate) were much higher. Higher concentrations could be due to pregnancy but it is probably reasonable to also look for causes in the type of sampling, sample preparation, and methods chosen. Furthermore, in the studies mentioned above, SCFAs were analyzed by gas chromatography, whereas in our study, we analyzed SCFAs by HPLC.

Instead of concentrations, the ratio between acetate, propionate, and butyrate (acetate–propionate–butyrate) is mentioned in the literature, which is about 60:20–25:15–20 in healthy people [14,55,56,57]. Jost et al. [51] determined a slightly different ratio between the three basic SCFAs in the feces of pregnant women, namely 65:15:20. Compared to studies with healthy, non-pregnant individuals, slightly more acetate and less propionate were measured. In our study, we obtained a similar ratio of 65:10:25. Compared to healthy people, the ratio differs mainly in propionate, which is present in lower concentrations in pregnant women [39,51] and pregnant women with GDM. Inulin propionate ester, which delivers propionate specifically to the large intestine, has been shown to reduce energy intake and prevent long-term weight gain in overweight adults. In addition, supplementation with inulin propionate significantly improved acute insulin secretion and β-cell function. This suggests that increasing colonic propionate improves glucose metabolism [58,59]. It is possible that the reduction in propionate in pregnant women is an adaptation of the gut microbiota to pregnancy, which allows for an increase in insulin resistance and ensures an adequate supply of nutrients to the fetus. It should be noted that SCFA concentrations in our study did not differ significantly between the trimesters of pregnancy.

Pregnant women with GDM and normal body weight before conception had significantly lower levels of acetate and propionate in their feces in both trimesters of pregnancy compared to pregnant women who were overweight or obese before conception. This is consistent with a number of studies that do not exclusively support a beneficial role for SCFAs. Metabolites of SCFA oxidation can be used for de novo lipid and glucose synthesis [44], and epidemiological studies in mice have shown that higher fecal SCFA concentrations are positively correlated with higher body weight, whereas lower SCFA values are associated with leanness [12,60]. Metagenomic studies have shown that in obese humans, the gut microbiome is enriched in pathways involving microbial fermentation of carbohydrates [46]. One possible explanation is that the obese gut microbiota leads to less efficient SCFA absorption and, therefore, increases SCFA excretion. The difference between fecal and serum SCFAs may provide an answer to this question, and studies using stable isotopes to measure SCFA dynamics would provide relevant data to address this [43,47]. A Colombian cohort showed that higher fecal SCFA concentrations were associated with indicators of lower gut microbiota diversity, higher gut permeability, systemic inflammation, glycemia, dyslipidemia, obesity, central obesity, and hypertension [47]. In our study, higher fecal SCFAs were also associated with higher blood pressure and with higher cholesterol and triglyceride levels. In the Colombian study, all associations persisted after adjustment for potential confounders such as dietary fiber, total calories, and physical activity, and the results were not explained by medication use [47].

Pregnant women with excessive weight gain during pregnancy had significantly higher butyrate concentrations in their feces in the third trimester compared to pregnant women with normal gestational weight gain (Figure 2b). Excessive weight gain during pregnancy increases the risk of a cesarean section, the birth of a macrosomic child, more difficult weight loss after birth, and also obesity in the child later in life [61,62,63]. Butyrate is described as an anti-inflammatory agent [64,65] and has been shown to influence the prevention of obesity and insulin resistance via the regulation of gut hormones [66,67]. Most of the microorganisms responsible for butyrate production are species such as *Roseburia* spp., *Ruminococcus* spp., *Clostridium* spp., *Anaerostipes* spp., *Butyrivibrio* spp., and *Coprococcus* spp. [65,68]. One study from a Mediterranean region of Northern Spain has shown a change (increase) in butyric acid from early to late pregnancy [69]. They suggested that increased concentrations could act as a countermeasure against the higher levels of inflammation that occur in late pregnancy compared to the first trimester. A change in the gut microbiota during pregnancy has been reported, associated with an increase in *Actinobacteria* and *Proteobacteria* and a decrease in *Faecalibacterium*, leading to gut inflammation and increased energy storage with maternal weight gain and hyperglycemia [35,70,71]. Although a decrease in *Faecalibacterium* in late pregnancy has been described, a slight increase in butyrate concentration has been found in pregnant women in late pregnancy [69]. The fluctuations in butyrate concentration during pregnancy do not appear to be related to the described changes in the microbiota but could be influenced by other characteristics such as the amount of microbiota present in the colon, the source of the substrate, intestinal transit time, host genetics, diet composition, and lifestyle factors [37,69,72]. Butyrate has received particular attention due to its positive effects on intestinal homeostasis but the role of butyrate in relation to weight gain and obesity remains controversial [65].

In our study, overweight and obese pregnant women gained significantly less weight during pregnancy than those who had a normal weight before pregnancy. Perhaps the overweight/obese pregnant women were additionally motivated by their participation in this study. The pregnant women were properly instructed and educated about a healthy diet [49], although patients and healthcare professionals were aware of the lack of care due to staff shortages in general in Slovenia [73], and additional education and information-sharing would probably have been in place. Our study has some limitations, including a small sample size, especially in the overweight/obese group. Although dietary patterns of women with GDM were analyzed [49], not enough emphasis was placed on monitoring fiber intake and other dietary components that affect SCFA levels. In addition, SCFAs in feces do not reflect serum levels and, in future studies, serum SCFAs should also be determined. Regardless of the conflicting publications on the role of SCFAs, our study showed that higher acetate and propionate levels were associated with pre-pregnancy obesity and higher butyrate levels were associated with excessive weight gain during pregnancy.

## 5. Conclusions

This study provides valuable insights into the relationship between fecal SCFAs and obesity and/or excessive gestational weight gain in women with GDM. We found that acetate and propionate concentrations in the feces were significantly higher in women with GDM who were overweight or obese before pregnancy compared to those with normal weight. However, butyrate concentrations did not show similar differences, except for an increase observed in the third trimester in women who experienced excessive weight gain during pregnancy. These findings suggest that SCFAs may be influenced by maternal body weight and weight gain patterns during pregnancy. While the role of SCFAs in obesity and GDM remains complex and not fully understood, our results highlight the potential for SCFAs, particularly acetate and propionate, to serve as biomarkers for maternal metabolic health. Further studies including larger sample sizes and additional measurements of serum SCFAs are needed to deepen our understanding of the underlying mechanisms and their implications for maternal and fetal health.

## Figures and Tables

**Figure 1 biomedicines-13-00387-f001:**
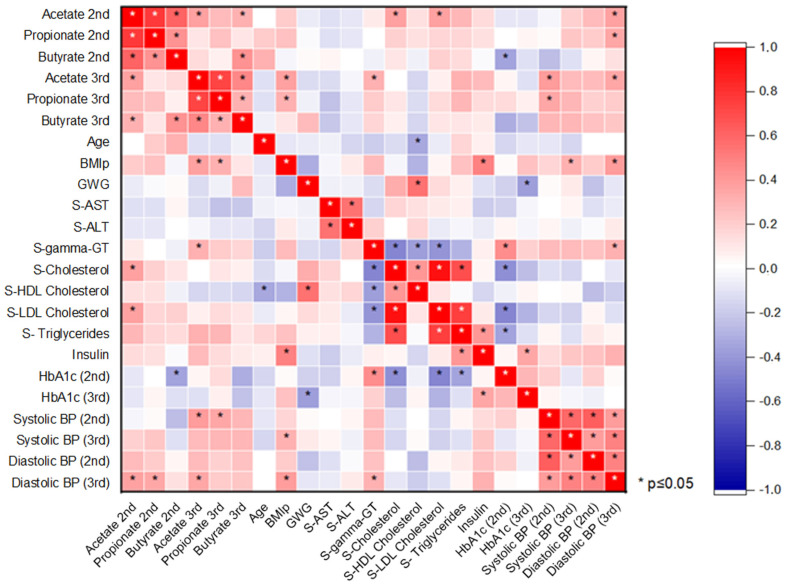
Pearson’s correlations between variables. The red color of a square indicates a positive correlation between the two variables and the blue color indicates a negative correlation. Statistically significant correlations (*p* ≤ 0.05) between variables are marked with an asterisk.

**Figure 2 biomedicines-13-00387-f002:**
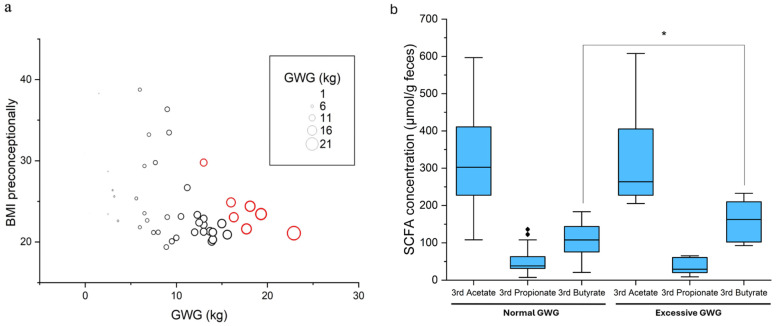
GWG and SCFAs in the 3rd trimester of pregnancy: (**a**) BMI preconceptionally and GWG. The red circles illustrate women with GDM who gained excessive weight during pregnancy; (**b**) SCFAs and GWG. Boxplots with 5-digit summary display (minimum, first quartile, median, third quartile, maximum). The diamonds illustrate the outliers. The asterisk indicates statistical significance at *p* < 0.05.

**Table 1 biomedicines-13-00387-t001:** Determined parameters of women with GDM included in this study. SD, standard deviation.

Variable	Mean (SD)/Median (25th, 75th Percentiles)			*p*-Value
	Total (*n* = 45)	Normal (*n* = 32)	Overweight/Obese (*n* = 13)	
Age (years)	32 (5)	32 (5)	31 (4)	0.961
BMI ^a^ preconceptionally (kg/m^2^)	23.0 (21.3, 27.2)	22.2 (21.2, 23.1)	29.8 (28.2, 34.9)	<0.001
GWG ^b^ (kg)	10.3 (6.7, 13.9)	12.4 (7.6, 15.5)	7.0 (2.5, 9.2)	0.017
Insulin (µU/l)–2nd trimester	9.4 (7.8)	7.7 (6.3)	13.9 (9.5)	0.002
Serum aspartate aminotransferase (S-AST)	0.27 (0.08)	0.25 (0.22, 0.32)	0.27 (0.19, 0.31)	0.797
Serum alanine transaminase (S-ALT)	0.26 (0.08)	0.23 (0.2, 0.32)	0.25 (0.23, 0.34)	0.402
Serum gamma-glutamyl transferase (S-gamma-GT)	0.14 (0.05)	0.11 (0.09, 0.15)	0.16 (0.14, 0.22)	0.006
S-cholesterol	6.5 (1.1)	6.7 (5.9, 7.3)	6.5 (5.7, 6.8)	0.402
S-HDL cholesterol	2.1 (0.4)	2.2 (2.0, 2.5)	2.0 (1.8, 2.2)	0.102
S-LDL cholesterol	3.6 (0.8)	3.7 (2.8, 4.3)	3.5 (3.0, 3.8)	0.859
S-triglycerides	1.8 (0.6)	1.8 (1.3, 2.0)	2.0 (1.7,2.3)	0.202
Hba1c ^c^–2nd trimester (%)	5.1 (0.4)	5.1 (0.3)	5.3 (0.5)	0.099
HbA1c ^c^–3rd trimester (%)	5.2 (0.3)	5.1 (0.4)	5.3 (0.3)	0.255
Systolic blood pressure (BP)–2nd trimester (mmHg)	117 (14)	113 (10)	124 (16)	0.057
Systolic blood pressure (BP)–3rd trimester (mmHg)	116 (17)	108 (13)	124 (21)	0.292
Diastolic blood pressure (BP)–2nd trimester (mmHg)	74 (9)	72 (7)	78 (9)	0.054
Diastolic blood pressure (BP)–3rd trimester (mmHg)	78 (12)	75 (13)	85 (11)	0.375

^a^ Body mass index. ^b^ Gestational weight gain. ^c^ Glycated hemoglobin.

**Table 2 biomedicines-13-00387-t002:** Acetate, propionate, and butyrate from feces of pregnant women with GDM in the 2nd and 3rd trimesters of pregnancy. SD, standard deviation.

SCFA	Mean (SD) (µmol/g Feces)			*p*-Value
	Total (*n* = 45)	Normal (*n* = 32)	Overweight/Obese (*n* = 13)	
Second Trimester	*n* = 43	*n* = 31	*n* = 12	
Acetate	315.7 (115.7)	289.4 (104.1)	383.2 (120.9)	0.043
Propionate	49.4 (21.5)	43.7 (19.4)	64.1 (20.3)	0.021
Butyrate	120.6 (51.7)	116.3 (41.3)	131.1 (72.9)	0.558
All SCFAs	482.8 (170.7)	457.9 (141.7)	578.5 (178.9)	0.044
Acetate–propionate–butyrate	65:10:25	64:10:26	66:11:23	
Third Trimester	*n* = 43	*n* = 30	*n* = 13	
Acetate	313.4 (128.6)	269.1 (92.3)	415.7 (145.3)	0.008
Propionate	48.7 (33.6)	39.4 (24.8)	70.1 (41.8)	0.029
Butyrate	117.3 (48.9)	111.4 (47.2)	131.1 (51.9)	0.444
All SCFAs	479.4 (183.1)	419.9 (138.1)	616.9 (204.9)	0.014
Acetate–propionate–butyrate	65:10:25	64:10:26	67:11:22	

## Data Availability

The data used and analyzed in the current study are available via Figshare, https://figshare.com/s/504c9f538f63c924a99b (accessed on 4 February 2025).

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
