# Peer review of "Fecal Short-Chain Fatty Acids Are Associated with Obesity in Gestational Diabetes"

_biomedicines, 2025, doi:10.3390/biomedicines13020387_

Round 1

Reviewer 1 Report

Comments and Suggestions for Authors

The manuscript entitled “Short-chain fatty acids in fecal samples from normal-weight and overweight/obese women with gestational diabetes” reports the changes in the SCFAs of healthy and obese women with gestational diabetes. The manuscript seems good, but we have to consider some of the concerns before processing the publication.

Major concerns:

·         Why do changes in SCFAs restrict the manuscript?

·         Why not the changes in microbiota?

·         Can the results be correlated with the general population?

Abstract: Fine.

Introduction:

·         The correlation between changes in SCFAs of healthy and obese persons needs to be included.

·         We must address the changes in gut microbiota and its role in diabetes, obesity, and gestational diabetes.

·         Also, supplements like probiotics and other fermented foods that benefit the obese and diabetic conditions are required. You may refer the papers like Role of Microbiome in obesity and Antiobesity Properties of Probiotic Supplements; Synbiotic supplementation improves obesity index and Metabolic Biomarkers in Thai obese adults, Influence of Bifidobacterium breve on the Glycaemic Control, Lipid Profile and Microbiome of Type 2 Diabetic Subjects.

Methods: fine

Discussion:

·         It needs more relevant recent findings, and the results must be correlated with other studies with geographical and ethnic variations in the study subjects.

·         How for we utilize the SCFAs as biomarkers to predict future metabolic issues in children like T2DM?

Conclusion:

·         A separate conclusion section is needed.

·         The study limitations should be highlighted in the conclusion.

Reviewer 2 Report

Comments and Suggestions for Authors

This study analyzed short-chain fatty acids in 45 women with gestational diabetes mellitus. Results showed higher acetate and propionate levels in overweight pregnant women, and elevated butyrate in those with excessive gestational weight gain. Some relationship between short chain fatty acids, obesity, and gestational diabetes mellitus was inferred.

This is a well-performed study, but lacks several information.

1 Physiological characteristics of pregnant women with GDM should be described in the introduction.

Comparison of fecal short chain fatty acids to other metabolic diseases should be discussed.

2 intestinal microbes that led to the increase of the short chain fatty acids should be discussed.

Reviewer 3 Report

Comments and Suggestions for Authors

It is interesting article about role of the short-chain fatty acids in the pathogenesis of obesity in pregnant women.

However, I did have a few comments.

Article

The title needs to be changed to one sentence and include the functional meaning of the article.

For example, “Diagnostic role of fecal short-chain fatty acids in the pathogenesis of obesity in pregnant women”.

Abstract

The abstract are well described. However, I would recommend adding a sentence at the end outlining the main takeaways of the article.

Introduction

The Introduction is short and should be including more references.

Materials and Methods

The Materials and Methods are well described.

Results

In Tables 1 and 2, it is better to use not percentiles, but SEM as the variance. It is better to show each of the studied samples separately on the diagram, given the small sample size of the experimental group (n=13). It is better to add this information as a separate figures.

The manuscript also lacks a general diagram showing the association between SCFAs and GDM.

Discussion

The discussion is very short. Interesting results were obtained and described, but the conclusions drawn are not very clear. It is mainly in the literature review. I recommend moving some of it to the Introduction. The discussion should clearly define all the key associations between obesity markers obtained during the experiment.

Conclusions

It is necessary to add a Conclusions chapter. I also do not recommend giving links in the conclusion.

References

References should include at least 50-60 references, most of which are from the last 5-6 years.

Reviewer 4 Report

Comments and Suggestions for Authors

This is well designed and written and I could not detect any typos. Some of the findings in the heat map about the relationships between the SFAs could be assessed in the discussion but I do not see that as a negative due to the complexity of the current understanding of the relationships

Round 2

Reviewer 1 Report

Comments and Suggestions for Authors

The manuscript has been improved after the revision 

Reviewer 3 Report

Comments and Suggestions for Authors

All corrections have been made by the authors. I have no more comments.